# Perspectives of Dietary Assessment in Human Health and Disease

**DOI:** 10.3390/nu14040830

**Published:** 2022-02-16

**Authors:** Aida Turrini

**Affiliations:** Independent Researcher, 58054 Scansano, Italy; aida.turrini@gmail.com

**Keywords:** dietary assessment, human health, disease, public health nutrition, nutritional-related concepts, nutritional databases, individual dietary survey, food consumption study, surveillance, nutritional database system

## Abstract

Diet and human health have a complex set of relationships, so it is crucial to identify the cause-effects paths and their management. Diet is crucial for maintaining health (prevention) and unhealthy diets or diet components can cause disease in the long term (non-communicable disease) but also in the short term (foodborne diseases). The present paper aims to provide a synthesis of current research in the field of dietary assessment in health and disease as an introduction to the special issue on “Dietary Assessment and Human Health and Disease”. Dietary assessment, continuously evolving in terms of methodology and tools, provides the core information basis for all the studies where it is necessary to disentangle the relationship between diet and human health and disease. Estimating dietary patterns allows for assessing dietary quality, adequacy, exposure, and environmental impact in nutritional surveillance so on the one hand, providing information for further clinical studies and on another hand, helping the policy to design tailored interventions considering individual and planetary health, considering that planetary health is crucial for individual health too, as the SARS-CoV-2 (COVID-19) pandemic has taught. Overall, dietary assessment should be a core component in One-Health-based initiatives to tackle public health nutrition issues.

## 1. Introduction

Diet is recognized as a core topic to deal with, then the Special Issue “Dietary assessment and human health and disease” was articulated by the editor including a set of cutting-edge topics: Direct effects the triple burden of malnutrition (under-nourishment, over-nourishment, and micronutrient deficiency); Potential effects/exposure: nutrients and undesirable food components (contaminants, residues, etc.) interactions; Indirect effects through environmental impacts: nutrients, food waste, and pollution/greenhouse gases emissions; Dietary assessment as an education tool; The landscape of dietary assessment used in different contexts for better-exploiting results: surveillance programs, epidemiological studies, and clinical trials; The role of dietary assessment in e-health and m-health applications [1].

The importance of the topic is emphasized by the number of sustainable development goals that are directly related to diet [2,3] by the European Food 2030 agenda [4], and last but not least by the constitution of the high-level panel of experts appointed by the European Commission [5], which is in line with other high-level expert groups set up by FAO [6] and the UN Food Summit [7,8].

Moreover, the centrality of the diet has been captured by many European research projects (from SUSFANS [9] to FIT4FOOD2030 [10]) and has been the subject of important works including the work produced by the EAT-Lancet expert commission [11] and a set of publications in complement of the arguments reported (see e.g., [12]). Classically, dietary guidelines representing the application at the public health nutrition level of scientific nutritional recommendations are designed to promote a healthy diet [13]. More recently, diet, health, and environment are also prominent in sustainable development goals (SDGs) [14], reflecting the recent tendency in including sustainability as an integral part of dietary guidelines [15]. In this context, many authors see the link between environmental protection and diet, and this is evidenced by the inclusion of food safety and sustainability in the guidelines for healthy eating [15,16].

Disentangling complex phenomena, like diet-health and disease, is a major challenge, as both human health and diseases have cause-effects interrelations with diet both at an acute and chronic level [17,18], are multi-directional, and are mediated by several factors, e.g., thinking on the diet-environment-health trilemma [19]. The main difficulty is represented by the complexity of each component, so a schematized vision can help to identify “critical control points” to start from in each specific situation.

Understanding the relationship between diet and health means analyzing several different links as is shown in Figure 1, it greatly relies on dietary assessment data, and there is a need «to encourage the development of improved methods and tools designed to assess and monitor diet and physical activity to help promote a healthier lifestyle» [20]. 

On the other hand, food security can affect health on a very different side related to hunger in the world. «The evidence also revealed that the world is not on track to achieve the SDG 2.1 Zero Hunger target by 2030. Combined projections of recent trends in the size and composition of the population, total food availability, and the degree of inequality in the food access point to an increase of the PoU by almost 1 percent. As a result, the global number of undernourished people in 2030 is projected to exceed 840 million» [21].

A huge amount of literature is devoted to studying all of these aspects. A simple exercise in searching keywords including “dietary assessment, human health, and disease” produced 3,230,000 results in Google Scholar. However, interestingly, when the exact wording was submitted, the result was “0”. A PubMed search found 21,636 titles (1 article was found by citation matching): For 1997, (Appendix A) applying the filters for that one year revealed 222 reviews (Appendix A), 200 “clinical trials” (Appendix A), 172 randomized control trials (Appendix A), 171 systematic reviews (Appendix A), 92 meta-analyses (Appendix A), and 2 “books and documents” (Appendix A). None of the titles were devoted to “dietary assessment, human health, and disease” per se, however, each publication considered one or more aspects in this field—dietary risks, sets of diseases, set of nutrients, population groups, dietary patterns, and many other topics.

The aim of the present work is to look at the contribution that dietary assessment can provide to disentangle the “diet-human health and disease” considering the context.

Figure 1 shows the mind-map [22] represented in a schematic way to look at the diet-health/disease relationships. Each entity (represented by rectangles) is in turn an outstanding topic that can be the object of research and/or worthy of being included in policy plans [23]. It evidences that some relationships are represented by direct links and some others are mediated. Lines represent relationships, and the arrows express the direction (e.g., how diet affects environmental safety through food and non-food wastes, which affects climate that affects food varieties and food components that can induce the modification of recommendations, etc.). These relationships are influenced/influence the food system (production, trade, food services, food environment, consumption) and can be managed by the public health nutrition system (surveillance, prevention, cure). The research provides input to support data-driven policy decision-making on food system sustainability and public health sector and infrastructures development, and modulate education and information programs. Diet and health have a direct link (blue double line double arrow line) but are also connected through different patterns. This vision inspires the following paragraphs where each piece of the rationalization is documented by scientific literature [24]. An unhealthy or unsafe diet either for under-, over-, or malnutrition can lead to non-communicable disease due to deficiencies, excesses, or unbalanced daily food intake [25]. However, these diets can influence non-communicable diseases through damage to the immune system [26].

## 2. Diet, Health, and Disease

Overall, the relationship between diet and health is complex mainly because both topics are complex in themselves but also because of the mediation of several other factors and the non-linearity of cause-effects [27]. 

It is known that the quality of the diet is fundamental for the maintenance of health [28] but diet deals with economic and social aspects that in turn can affect human health [29]. From a short-term perspective, poverty can be associated with an environment that can bring unsafe substances into the diet and then, potential food-borne disease [30]. In the long-term, poor nutrition knowledge can be associated with the adoption of a non-balanced diet and then, possible non-communicable diseases [31,32]. 

The economic situation can crucially affect the quality of the diet at both the individual and community levels [33]. Food purchase choices by consumers contribute to defining the food market and, therefore, have a return effect on the world of food production, quality of food (including diversity), and, again, the diet [33]. The cultural model includes traditions, information and education, and individual attitudes that affect eating patterns, diet quality, human health, environmental impact, and so on (see e.g., [34]). 

The supply chain has effects on the diet through the food quality—nutrients, beneficial non-nutrients, harmful substances, —and the impact on health can be either in the long-term for substances chronically ingested or in the short term essentially for contamination with immediate effects (microbiological contaminations, poison, and other foodborne diseases) [35]. 

For example, food consumption-related activities can produce an environmental impact that contributes to climate changes and other detrimental effects on the environment [33], yet at the same time, a degraded environment has in turn effects on human health directly or through impacts on food production, the nutritional composition of food [36], and, therefore, on the diet. 

Diet, e.g., through greenhouse gas emissions related to consumed foods, has impacts on the climate [37,38] and environment [39,40] through food choices influencing the food market and then the food system (production, processing, delivering]. Climate and environment in turn have effects on both food quality [41,42] and directly on health [43]. 

In general, we can observe that on the research side, diet and disease are investigated to prevent disease onset, and clinical trials and surveillance are carried out to maintain human health and to adopt sustainable diets for planetary health [44,45]. 

The main difference between dietary factors causing non-communicable diseases and foodborne diseases is that non-communicable diseases, especially obesity, are mainly related to unhealthy dietary patterns adopted in the long term [46,47]. In contrast, spot adverse events causing foodborne disease are mainly related to specific eating occasions where foods are consumed that carry harmful substances [48]. 

However, disease status can have effects on diet as it requires dietary adjustments [49].

Once the disease is overt, the diet represents an element of knowledge in the anamnesis of the patient, or can be is used as a therapeutical instrument, and the disease can strongly condition the diet to follow.

The diet and disease relationship in a therapeutic setting is considered in a two-way direction. On the one hand, the dietary history can help to understand the cause that has led to the disease [50]. On the other hand, a diet therapy plan must be developed to cure the disease or adjust the current diet to the disease status [51].

Recommendations on diet, chronic diseases, and health highlight the role of diet in preventing chronic disease, particularly reducing dietary risk factors [52]. Multidimensional approaches are important in the management of chronic conditions, but nutritional interventions are critical and central to these strategies [49].

Overall, we can say that diet, health, and disease links have a multidimensional structure that is multi-directional. To disentangle such complex relationships, dietary assessment must provide a set of information that can be used in different sectors.

## 3. Dietary Assessment

“Nutritional assessment can be defined as the interpretation from dietary, laboratory, anthropometric, and clinical studies. It is used to determine the nutritional status of individual or population groups as influenced by the intake and utilization of nutrients” [53,54].

At the public health nutrition level, the dietary assessment allows for estimating the quality of the diet adopted by the population in terms of the evaluation of dietary patterns (e.g., the Healthy Eating Index [55]), adherence to a healthy dietary pattern (e.g., the Mediterranean Diet [56,57]), the dietary exposure to beneficial and harmful substances intake within the risk assessment process [58], and, last but not least, the environmental impact of dietary habits [59]. Nutrition epidemiological studies allow for estimating the association of all these variables to disease [60,61]. 

Looking at the research side, diet can be considered an experimental factor or a treatment in clinical trials and interventions studies, or an objective in estimating dietary patterns per se or as part of epidemiological studies [61]. Dietary assessment is a pillar in clinical trials [62] and dietary biomarkers studies [63,64], including calibration for the reliability of methods [65] and nutritional interventions, including education and information [66], and population studies (individual food consumption surveys [67,68], nutritional epidemiology [69,70], nutritional surveillance [71]), other than the anamnesis of patients in clinical settings [72,73,74] and clients in nutritionists’ consultancy and shaping personal nutrition plans or products [75].

A dietary assessment provides a core information system and requires a composite nutritional information system whose components can greatly help to identify crucial points in the “diet, health, and disease” relation [76]. All the information needs to be elaborated to become useful data. Underpinning data are necessary to perform this task, particularly instructions for coding data (to estimating under/over-reporting, codification for age class, body mass index, etc.), food coding system(s) to link food intake data to servings databases for quantification, thresholds for lower/upper limits for nutrients, acceptable/tolerable/permitted levels for contaminants/residues/other harmful substances occurrences databases, food composition databases, and greenhouse gas emission databases.

The core information describes amounts of food and nutrients, either actual or habitual intakes [77], the potential dietary exposure to chemical hazards over short (acute) or long periods in time (chronic) [78,79,80,81], environmental impacts, and, in the case of some specific methods (like e.g., precise weighing), the evaluation of left-overs is also performed [82]. The system can be used in evaluating the dietary patterns and estimating the nutritional status in population studies and analyzing the relationship between diet and health and disease [82] or evaluating the patients in a clinical context to obtain a complete assessment examining multiple components [83]. 

Such information can be collected in different types of settings: population studies (dietary surveys, surveillance programs (systematic collections), epidemiological studies (cohort study, case-control study, cross-sectional surveys), intervention studies, clinical trials, and individual treatments including disease treatments, nutrition consultancy, and personalized nutrition [84]. Data providing proxy of food consumption with the main important characteristics to provide a time series of food supply (food balance sheets—FBS) and household food availability in household budget surveys (HBS) both used in estimating trends in the population [85,86]. Precision and time references vary across these contexts. Therefore, the dietary assessment technique varies according to study aims, circumstances, and resources [87,88]. Therefore, dietary studies can be conducted on a large or a small scale according to the aim and the required food intake measurement: the large-scale studies allow the arrangement and maintenance of large information bases for several applications, while small-scale studies are usually devoted to deepening the knowledge on particular aspects, e.g., specific population groups (special diets, type of workers, etc.), specific substances (nutrients, contaminants, additives), or specific environments (institutionalized people, out-of-home food services, coastal areas, etc.), validation studies, intervention studies, and high precision measurement. It is known that the deeper the study, the greater the interview burden and the lower the response rate, so a compromise is necessary especially if resources are limited [87,88].

Dietary intakes cannot be observed per se [82] because what can be observed is the amount “on the plate” and the left-over amount to estimate the eaten amount. This operation can be concretely done by means of scales [82] or visually evaluated compared with prototypal servings (the food photos atlas) [89] or quantified according to standard units of measure [90]. A plethora of methodologies have been developed to quantify actual (prospective) or usual (retrospective) eating habits [78,79,80,81], each one responding to the adoption of different perspectives defined by the purpose of the study, the planned analyses, and the available resources [91], taking also into account the need for validation necessary when a new method is introduced [92]. A typical case is the adaptation of a food frequency questionnaire to reflect food habits in different population groups (see e.g., [93]). The availability of new tools allows for enhancing the survey techniques developing new approaches [76] reducing the commitment because in general, the less burdensome the procedure, the more selected individuals will participate in the studies [61,94,95].

Absolute precision is an impossible goal [91] in this field because of the nature of the issue; therefore, data usability must be guaranteed by fixing some principles, in particular: well-defining all steps necessary to carry out the dietary study; clear methodology in collecting food intake data; harmonization of survey methods, particularly the steps (study plan, sampling design, logistic), tools (software, algorithms, classification systems), and procedures (data processing); completeness and reliability of underpinning databases (food products, portion sizes, recipes; food composition data), and; appropriateness of statistics according to the sampling design and the objective of the study. 

In general, different kinds of studies provide information that can be mutually exploited [96]. As an example, comparing dietary patterns and determinants across population groups allows for highlighting current and emerging topics. This information can be used by researchers in designing and conducting clinical trials and intervention studies. On the other hand, clinical trial research can be exploited in planning new dietary surveys and using more appropriate new tools (e-health, m-health) [97,98,99]. Database management is always challenging, but collaboration between research groups and among citizens can improve the whole information system, allowing for the underpinning of policy-making decisions.

Dietary assessment is a continuously evolving issue from the individual level to the planetary one. 

Contributions to the special issue offer insights that identify dietary assessment as a core topic to develop according to the several challenges presented above.

Here, Table 1 summarizes the topics handled by the authors. We mostly found papers concerning the data and methodological aspects: dietary measurement methods (*n* = 5), data analysis approaches (*n* = 1), and nutritional database (*n* = 1). The second set of papers covers diet and health (outcomes, parameters, healthcare) (*n* = 3). Finally, papers investigating nutrition knowledge in the professional sector are presented (*n* = 2). Overall, the articles published in the “Dietary Assessment and Human Health and Disease” special issue represent the breadth and the state-of-the-art in this field.

The range of published articles does not cover all topics considering the large number of themes implied by the diet-health relationship, but it indicates that there are areas not yet completely developed in the current literature. Among these are dietary assessment in the clinical context, dietary assessment and health outcomes, and so on. 

## 4. Future Perspectives

Enhancing technological tools is always important [76] with consequences on the governance of complex nutritional database systems (e.g., the food composition data to confer reliability to the tips provided by apps on smartphones) [100]. Another increasingly important issue is the possibility to integrate dietary assessment and other techniques belonging to metabolomics [101] in general, and biomarkers studies [102], that are proposed for “objective dietary assessment” [63].

Several data are available but a “data deluge” can be equivalent to a “data desert” if an adequate knowledge of their nature is not complete [103]. Misusing the information is always a possibility but infrastructure providing metadata, focal points, and expert contact persons will greatly help manage these issues.

There is a vast space for research and policy to look at all of them and assess the multiple sectors involved and their interconnections in a multi- inter-transdisciplinary way [23,66,104,105].

As the food system and public health interact with each of the components [106], the One-Health approach initially born to manage food safety issues can be applied to manage all aspects [107]. The WHO defines One Health as “an approach to designing and implementing programs, policies, legislation, and research in which multiple sectors communicate and work together to achieve better public health outcomes” [108,109]. It looks at all sources of danger involving actors that can act to present or manage factors limiting these problems [110]. The One-Health [111] concept and the Healthy and Sustainable Diet framework are expected to guide research and policy in planning research programs, infrastructures, health systems, and interventions to achieve sustainable development goals and prevent future pandemics. Then, monitoring dietary patterns should be performed at regular time intervals to support the decision-making process. This will also help in evaluating a component of exposome where ecosystem, social, and lifestyle interact with physical and chemical aspects [112]. Advancement in self-assessment web or mobile device or wearable tools will greatly help in performing dietary assessment [113].

## Figures and Tables

**Figure 1 nutrients-14-00830-f001:**
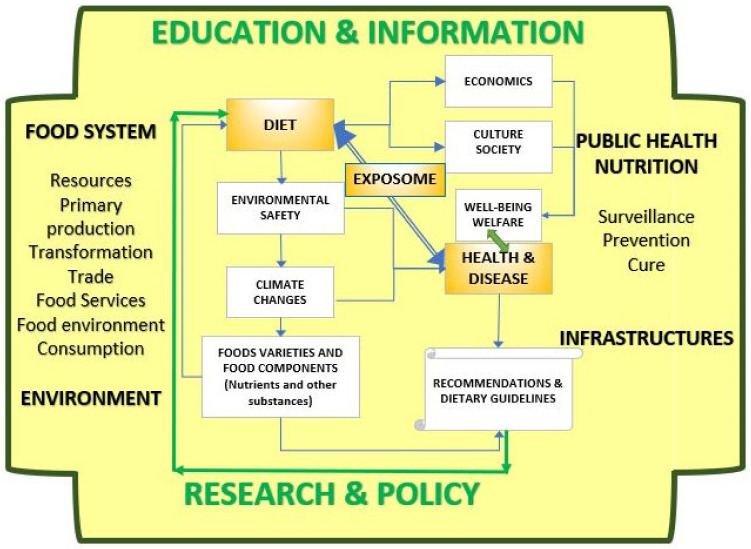
Diet health and disease: a visual map connecting components. Source: own design. Note: each rectangle represents a concept the lines indicate direct links and the arrows indicate the direction. Diet-health and disease relationships are regulated by the exposome, condensing determinants coming from the diverse components, and are immersed in a wider system whose major components are food system and public health nutrition formulating policies, supporting research, planning education, and information programs.

**Table 1 nutrients-14-00830-t001:** Special issue “Dietary Assessment and Human Health and Disease” titles of publications.

Topic	Title
Dietary measurement method	Validity and Reproducibility of a Culture-Specific Food Frequency Questionnaire in Lebanon
	Exploring the Validity of the 14-Item Mediterranean Diet Adherence Screener (MEDAS): A Cross-National Study in Seven European Countries around the Mediterranean Region
	Development and Validation of a Questionnaire to Assess Adherence to the Healthy Food Pyramid in Spanish Adults
	Development, Relative Validity and Reproducibility of the Aus-SDS (Australian Short Dietary Screener) in Adults Aged 70 Years and above
	Comparison of Self-Reported Speed of Eating with an Objective Measure of Eating Rate
Data analysis approaches	Exploration of the Principal Component Analysis (PCA) Approach in Synthesizing the Diet Quality of the Malaysian Population
Nutritional databases	Extractable and Non-Extractable Antioxidants Composition in the eBASIS Database: A Key Tool for Dietary Assessment in Human Health and Disease Research
Diet and health outcomes	Is Caloric Restriction Associated with Better Healthy Aging Outcomes? A Systematic Review and Meta-Analysis of Randomized Controlled Trials
	Hemodialysis—Nutritional Flaws in Diagnosis and Prescriptions. Could Amino Acid Losses Be the Sharpest “Sword of Damocles”?
	Celiac Dietary Adherence Test and Standardized Dietician Evaluation in Assessment of Adherence to a Gluten-Free Diet in Patients with Celiac Disease
Nutrition knowledge	Effectiveness of Diet Habits and Active Life in Vocational Training for Higher Technician in Dietetics: Contrast between the Traditional Method and the Digital Resources
	What Healthcare Professionals Think of “Nutrition & Diet” Apps: An International Survey

Note: Each title has a hyperlink to the open-access publication.

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
