# Peer review of "Perspectives of Dietary Assessment in Human Health and Disease"

_nutrients, 2022, doi:10.3390/nu14040830_

Round 1

Reviewer 1 Report

The manuscript presents an introduction to the special issue. It gives the reader essential information about what will be present in this issue. It is well written, in a logical manner, and brings welcomed topics as One health, food systems and the role of dietary assessment.

As minor suggestions, please add the source in figure 1.

Lines 80-82, maybe would be interesting consider also the diseases due to deficient intake, as anemia.

The lines 117-119 repeat the sentence on lines  80-82. Consider rewriting this part.

Reviewer 2 Report

The review is a very extensive overview of the entire area Diet and Health. As such it broadly covers a wide range of topics and definitions but the downside is that it only superficially introduces each of these key concepts, covering each with definitions originating from 1-2 published articles cited per concept. There is no depth in the discussion of the concepts, no summary of the latest evidence from systematic reviews and meta-analyses, and no critical reflection of the area. Hence, it is difficult to see what the contribution is of this review to the existing literature.

Mostly, the language is fine but in places it needs revision.

Introduction: Why is it important to list what is included as the set of articles in the Special Issue? Has this been requested by the journal?

Figure 1: does this come straight out of reference 20, or has it been created by the author? This is unclear.

Some references seem unnecessary, for example number 22 that describes what a mind-map is.

What is the value of Table 1? Has this been requested by the journal? Is this review an introduction to a special issue? Otherwise I don’t see any reason for Table 1.

Reviewer 3 Report

Thank you for the opportunity to review this paper.  It covers a really interesting topic of great relevance to all working in diet, nutrition and health.  However the aims are currently too broad and would benefit from significant reshaping to help direct the structure of the paper.

Please find specific suggestions below.  This is not an exhaustive list but highlights some of the common issues which would benefit from attention.

General

I would recommend a detailed review of the English throughout to improve clarity and allow the arguments to be more clearly made.

Please ensure any abbreviations are written in full before you abbreviate.

Abstract:

'Diet is crucial for maintaining a healthy status' – consider rephrasing as ‘diet is crucial for maintaining health’

Line 10 – ‘The dietary assessment’ – remove ‘the’

There are a few vague statements in the abstract which are hard to follow without reading the full paper – consider rephrasing, e.g. ‘provides the core information basis for several studies in all the different contexts’ and ‘Dietary assessment in small-scale dietary surveys allows for deepening specific aspects according to the aspects above mentioned’

Line 17 – ‘A dietary surveillance system building and maintaining a comprehensive database system would provide suitable information to assess diet, health, and disease relationships at the population level.’ – not sure what this means, consider rephrasing

Could the abstract introduce the history of dietary assessment – give the reader some context as to how long assessments have been in place and why you are revisiting this now, i.e. it is not a new issue and the reasons for needing DA are not new but perhaps methods have changed/ improved, diets are more complex etc.  As it stands the abstract does not currently explain why this topic is being addressed now.

Intro

Who articulated the topics you mention? Did they come from a workshop/ survey? Are they the authors opinion?

Line 44 – rephrase ‘a core of publications in complement of the arguments reported’ – ‘supporting the arguments reported’

Line 54 – ‘are in a multi-directional way mode,’  should this just be ‘are multi-directional’

Fig 1 – there are a number of spelling errors in the figure, the difference between direct and mediating relationships (see later comment) is not clear

Line 73 -  “diet”-“human health and disease” relationship. – remove the “”

'It evidences that some relations are represented by direct links some others are mediated'. – this is not clear from the diagram – the relationships between the different entities need to be more clearly distinguished – all arrows appear the same currently.

Line 81 – replace the term ‘follow-up’

Line 91 – ‘In the long-term poor nutrition, knowledge’ – should this be ‘and knowledge’?  Doesn’t make sense currently

Generally too many sweeping statements with insufficient explanation, e.g. Section diet and health – ‘Diet has impacts on climate [37,38] and environment [39,40] through food choice influencing food market and then the food system (production, processing, delivering]. Climate and environment in turn have effects on both food quality [41,42] and also directly on health [43]. – each of these needs examples – how does climate affect food quality? How does this affect dietary assessment?  It might make sense to narrow down the range of topics and focus on those more closely linked to choice of dietary assessments and/ or developments in dietary assessment.  Currently there seems to be too many topics covered in too little depth.

In section 2 – Diet & health you cover diseases linked to diet yet have a separate section (section 3) on disease and diet. – can health and disease be logically separated when they are likely to be on the same continuum?  Perhaps reconsider your section headings and structure.

Line 117 – this duplicates line 81

Line 120 – ‘and risks on acute’ – what does this mean?

Line 122 - while spot adverse events are mainly related to specific eating occasions consuming foods carrying in unsafe substances' - what does this mean?

Line 130 – the diet can be parsed to understand the cause that has determined 130 the disease in a retrospective view‘ – what does this mean?

Line 133 – ‘on a diet’ remove the ‘a’

Line 144 – how can a direct quote have 2 refs assigned to it?

Line 146 – remove ‘synthetic’

Line 157 – where is the close bracket?

Line 176 – ‘in the curing phase’ – what does this mean?

Line 177-184 –  this duplicates line 170-177 – whole paragraph is repeated

Line 188 – ‘and individual cares’ – what does this mean?

Line 216 – ‘less invasive’ – agree but could you not have a non-invasive technique that is still high burden? Most methods, other than biomarkers are not highly invasive but still place a lot of burden on the participants. Would it be clearer to talk about burden?

Line 220 - dietary study phases – what does this mean?

Line 223 – ‘A general consideration about mutual exploitation of information comes from different kinds of studies’ – what does this mean?

Line 237 – ‘Other data processing and underpinning databases’ - this is not a complete sentence

Line 255 – ‘always behind the corner’ – what does this mean – should this be ‘always around the corner’?

Round 2

Reviewer 2 Report

I made my points in my first review and these are still valid. I leave to the editor to make the final decision.

Reviewer 3 Report

Thank you for taking onboard the previous comments. Some further suggestions or comments are listed below

General

Although improvements have been made a thorough review of the English is still warranted to clarify a number of the statements made.  Errors in English are NOT listed here.

Abstract

The present paper aims to provide a synthetic view of the topics that research in dietary assessment and human health and disease includes. The special issue is aimed to collect papers highlighting “Dietary Assessment and Human Health and Disease” - consider rephrasing as ‘a synthesis of current research in the field of dietary assessment in health and disease as an introduction to the special issue on….’

Line 15 – replace several with something ‘bigger’!, e.g. many, a host of… etc. Several sounds far too few for such a vast field of research.

'Overall, dietary assessment seems suitable as a core component in a One-health approach to tackling public health nutrition issues'. – suggest: 'overall dietary assessment should be a core component ….' (needs to be stronger)

Intro

Line 77 – the full text of the abbreviation should just be written in the sentence rather than as a footnote.  I do not believe this is used again so there is no need to abbreviate.

Figure 1 – this needs it’s own footnote/ legend explaining what the different line styles represent – I appreciate this is provided later in the main text but figures should be able to stand alone

Line 232 - 'therefore, data usability must be guaranteed by fixing some principles, particularly: well-defining all steps necessary to carry out the dietary study phases; Methodology in collecting food intake data; harmonization of survey methods; procedures of dietary assessment; standardization of procedures application; information system completeness and reliability; appropriateness of statistics'. – there seems to be some duplication in this list – not clear how, for example, 'methodology in collecting food intake data' and 'procedures of dietary assessment' are different.

Line 251 – Would suggest rephrasing as shown to improve clarity: 'Here, table 1 summarizes the topics treated handled by the Authors. we find mostly papers concerning both the data and methodological side aspects: dietary measurement methods (n=5), data analysis approaches (n=1), nutritional database (n=1). A second set of papers cover diet & health (outcomes, parameters, healthcare) (n=3). Finally, papers investigating nutrition knowledge in the professional sector are presented (n=2). Overall, the articles published in the this “Dietary Assessment and Human Health and Disease” special issue represent a sample of several that reflects the breadth and the state of the art in this field.'

Line 259 - 'The range of published articles does not cover all topics considering the large number of themes implied by the diet-health relationship, but indicates areas for future development this is a clue to look at also to evaluate which issues can be further developed like dietary assessment in the clinical context, reviews on dietary assessment and health outcomes, and so on.'

Please explain what is meant by: ‘review on dietary assessment’ – could this be made more specific, e.g. reviews on use of technology, techniques to improve accuracy, techniques to reduce burden?  What specifically needs to be reviewed?

Line 270 – ‘Achieving Sustainable Development Goals, facing pandemics poses the issue to empower tools and communities correctly using these’. – I am not sure what this means – please consider rephrasing

Line 285 – remove full stop

Line 294 – not sure I follow this final sentence: ‘The dietary assessment is then expected to be performed at a regular time interval to support the decision-making process’ – dietary assessment of who?
